# UAV3D: A Large-scale 3D Perception Benchmark for Unmanned Aerial Vehicles

**Hui Ye**
Dept. of Computer Science
Georgia State University
Atlanta, GA 30303
hye2@student.gsu.edu

**Rajshekhar Sunderraman**
Dept. of Computer Science
Georgia State University
Atlanta, GA 30303
rsunderraman@gsu.edu

**Shihao Ji**[*]
School of Computing
University of Connecticut
Storrs, CT 06269
shihao.ji@uconn.edu

## Abstract

Unmanned Aerial Vehicles (UAVs), equipped with cameras, are employed in numerous applications, including aerial photography, surveillance, and agriculture. In these applications, robust object detection and tracking are essential for the effective deployment of UAVs. However, existing benchmarks for UAV applications are mainly designed for traditional 2D perception tasks, restricting the development of real-world applications that require a 3D understanding of the environment. Furthermore, despite recent advancements in single-UAV perception, limited views of a single UAV platform significantly constrain its perception capabilities over long distances or in occluded areas. To address these challenges, we introduce UAV3D – a benchmark designed to advance research in both 3D and collaborative 3D perception tasks with UAVs. UAV3D comprises 1,000 scenes, each of which has 20 frames with fully annotated 3D bounding boxes on vehicles. We provide the benchmark for four 3D perception tasks: single-UAV 3D object detection, single-UAV object tracking, collaborative-UAV 3D object detection, and collaborative-UAV object tracking. Our dataset and code are available at https://huiyegit.github.io/UAV3D_Benchmark/.

## 1 Introduction

Due to their superior mobility, Unmanned Aerial Vehicles (UAVs) are widely deployed and crucially important in advancing numerous applications of computer vision, such as traffic monitoring, precision agriculture, disaster management, and wildlife surveillance, offering more efficiency and adaptability than traditional surveillance cameras with fixed perspectives. On the other hand, the availability of large-scale public datasets and benchmarks, such as ImageNet [2] and MSCOCO [11], has significantly accelerated progress of various computer vision tasks, including image classification, object detection, object tracking, semantic segmentation, and instance segmentation.

3D perception plays an essential role in vision-based robotic systems, enabling them to handle complex tasks beyond the capabilities of 2D perception. While still a relatively new technology for UAVs, 3D vision offers the ability to capture complete dimensional data of objects in a 3D environment. Currently, most UAV datasets [4, 31] are mainly designed for 2D perception, limiting the development of real-world applications that require 3D understanding of the surrounding environment. To bridge this gap, we introduce UAV3D to advance the research in 3D perception with UAVs.

Furthermore, despite recent advancements in single-UAV perception, limited views of a single UAV platform significantly constrain its perception capabilities over long distances or in occluded areas. A promising solution lies in UAV-to-UAV communication, a cutting-edge technology that facilitates

---

[*]Part of the work was done while the author was affiliated with Georgia State University.

38th Conference on Neural Information Processing Systems (NeurIPS 2024) Track on Datasets and Benchmarks.

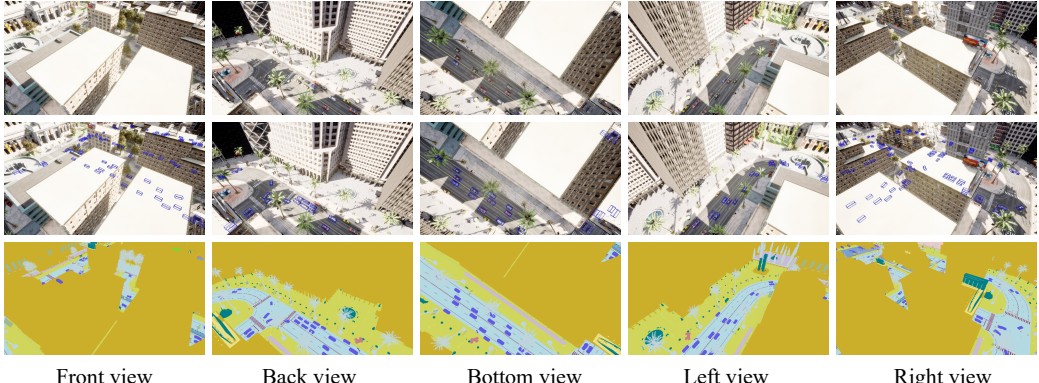

| Front view | Back view | Bottom view | Left view | Right view |

Figure 1: An example from the UAV3D dataset. From the top to bottom, they are 5 different RGB images from different camera views, images with 3D bounding boxes, and images with semantic labels.

collaborative information exchange among UAVs. While there are several recent collaborative perception datasets [5, 27, 30, 8, 29] for autonomous driving, there remains a noticeable gap of well-developed and organized collaborative perception datasets specifically for UAVs. Operating multiple UAVs simultaneously to build such a dataset is extremely challenging due to the high costs and extensive labor involved. Therefore, we focus on the development of a simulated dataset UAV3D to advance the research of collaborative perception for UAVs.

To create the UAV3D dataset, we utilize CARLA [3], a popular open-source simulator for autonomous driving, along with AirSim [21] to simulate flying UAVs. The camera sensors mounted on UAVs are recorded synchronously to facilitate collaborative perception. In addition, diverse annotations, including bounding boxes, vehicle trajectories, and semantic labels are provided to support various downstream tasks. Figure 1 shows an example from the UAV3D dataset. To better support multi-UAV and multi-task perception research for UAVs, we further provide the benchmarks for four 3D perception tasks, including single-UAV 3D object detection, single-UAV object tracking, collaborative-UAV 3D object detection, and collaborative-UAV object tracking.

## 2 Related work

Over the past few years, several datasets have been developed to advance the perception tasks in autonomous driving and UAVs. The UAVDT [4] dataset comprises 100 video sequences selected from more than 10 hours of footage captured by a UAV platform across various urban locations. The VisDrone [31] dataset includes 263 video clips, comprising 179k frames and 10k static images, captured by diverse drone-mounted cameras from multiple locations and weather conditions. However, both benchmarks are small-scale, real-world datasets designed for 2D perception tasks. In contrast, our UAV3D is a large-scale simulated dataset, comprising 500k images for 3D perception tasks. The Waymo Open [22] and nuScenes [1] datasets are two public, large-scale, multi-modal datasets that include camera, radar, and lidar data for autonomous driving. Our UAV3D comprises 1,000 scenes, matching the scale of the nuScenes dataset. It also has the same format as nuScenes to provide annotations and metadata, such as calibration, maps, and vehicle coordinates. OPV2V [29], V2X-Sim [8], and V2XSet [28] are three simulated multi-agent perception datasets designed for V2X-aided autonomous driving. OPV2V supports multi-modal data and multi-agent collaborative perception in a vehicle-to-vehicle scenario. Meanwhile, V2X-Sim and V2XSet provide multi-agent sensor recordings from roadside units (RSUs) and multiple vehicles, enabling collaborative perception in both vehicle-to-vehicle and vehicle-to-roadside scenarios. Our UAV3D utilizes the same baseline models as V2X-Sim to evaluate the collaborative perception tasks for UAVs. CoPerception-UAV [6] is a simulated dataset for UAV-based collaborative perception, featuring 131.9k synchronous images captured at three different altitudes across three towns and in two swarm formations. In comparison, our UAV3D contains 500k images captured over four diverse towns, making it approximately four times larger in scale than CoPerception-UAV. DAIR-V2X [30], V2V4Real [27], Rcooper [5], TUMTraf-V2X [32], HoloVIC [20] and V2X-Real [26] are six large-scale, real-world datasets that facilitate vehicle-centric

Table 1: Comparison of recent related datasets for autonomous driving and UAVs. Real and Sim refer to real-world and simulated dataset, respectively. V2X, V2V and V2I denote the vehicle-to-everything, vehicle-to-vehicle and vehicle-to-infrastructure cooperation, respectively. C, L and R refer to Camera, Lidar and Radar sensor, respectively. While CoPerception-UAV includes vehicles with 21 different categories, UAV3D has 17 different categories of vehicles.

| Dataset | Year | Real/sim | Scenario | V2X | Modality | Scenes | Frames | Images | 3D boxes | Classes |
|---------|------|----------|----------|-----|----------|--------|--------|--------|----------|---------|
| VisDrone [31] | 2018 | Real | Drone | No | C | – | 179.2K | 10.2K | No | 10 |
| UAVDT [4] | 2018 | Real | Drone | No | C | – | 80K | 80K | No | 3 |
| Waymo Open [22] | 2019 | Real | Driving | No | C & L | 1K | 200K | 1M | 12M | 4 |
| nuScenes [1] | 2019 | Real | Driving | No | C & L & R | 1K | 40K | 1.4M | 1.4M | 23 |
| OPV2V [29] | 2022 | Sim | Driving | V2V | C & L | – | – | 44K | 230K | 1 |
| V2X-Sim [8] | 2022 | Sim | Driving | V2X | C & L | – | – | 60K | 26.6K | 1 |
| V2XSet [28] | 2022 | Sim | Driving | V2X | C & L | – | – | 44K | 230K | 1 |
| DAIR-V2X [30] | 2022 | Real | Driving | V2I | C & L | – | – | 39K | 464K | 10 |
| CoPerception-UAV [6] | 2022 | Sim | Drone | V2V | C | 183 | 4.4K | 131.9K | 1.6M | 21 |
| V2V4Real [27] | 2023 | Real | Driving | V2V | C & L | – | – | 40K | 240K | 5 |
| Rcooper [5] | 2024 | Real | Driving | V2I | C & L | – | – | 50K | – | 10 |
| TUMTraf-V2X [32] | 2024 | Real | Driving | V2I | C & L | – | – | 5K | 29.3K | 8 |
| HoloVIC [20] | 2024 | Real | Driving | V2I | C & L | – | 100K | – | 11.4M | 3 |
| V2X-Real [26] | 2024 | Real | Driving | V2X | C & L | – | – | 171K | 1.2M | 10 |
| UAV3D | 2024 | Sim | Drone | V2V | C | 1K | 20K | 500K | 3.3M | 17 |

collaborative perception for autonomous driving, each emphasizing different cooperative scenarios: vehicle-to-infrastructure cooperation in DAIR-V2X, vehicle-to-vehicle cooperation in V2V4Real, vehicle-to-roadside cooperation in Rcooper, vehicle-to-infrastructure cooperation in TUMTraf-V2X, vehicle-to-infrastructure cooperation in HoloVIC, vehicle-to-vehicle and vehicle-to-infrastructure cooperation in V2X-Real. In contrast, UAV3D is a large-scale simulation dataset designed to support UAV-centric collaborative perception, specifically focusing on UAV-to-UAV cooperation scenario. The comparison of related datasets is summarized in Table 1.

## 3 Benchmark Dataset

### 3.1 Data Collection

**CARLA-AirSim Co-simulation.** We utilize the open-source CARLA [3] and AirSim [21] simulators for traffic flow simulation and data recording. In CARLA, vehicles are spawned to navigate randomly throughout the town. Hundreds of vehicles (i.e., 200) are active in each town (Towns 3, 6, 7, and 10), which feature complex traffic situations, such as crossroads and T-junctions. We record 250 log files per town, resulting in a total of 1,000 scenes.

**Flight Planning.** We operate UAVs in Towns 3, 6, 7, and 10 of CARLA, with Town 10 being particularly known for its dense traffic and highly challenging driving situations. We emphasize the variations between urban (Towns 3 and 10) and suburban (Towns 6 and 7) settings, particularly in terms of traffic flow, vegetation, architecture, vehicles, and road markings. For each town in CARLA, we have established 25 flight routes to cover a diverse range of locations from the bottom left to the top right of the map.

**Sensor Setup.** As shown in Figure 2, we equip each UAV with five RGB cameras to capture both RGB and semantic images. Four of these cameras face the front, left, right, and back with a pitch angle of -45 degree, while the bottom camera provides a bird's eye view. The resolution of the images is 800x450 pixels. CARLA uses the Unreal Engine (UE) coordinate system, with x-forward, y-right, and z-up, returning coordinates in local space. AirSim, on the other hand, adopts the North-East-Down (NED) coordinate system, where north aligns with the Unreal Engine x-axis. We adjust the sensor coordinates from AirSim to align with vehicle coordinates from CARLA.

**UAV-Swarm Formation.** As shown in Figure 3, we configure a swarm of five UAVs in a cross-shaped formation with the positions at the front, left, right, center, and back, each with a 20-meter distance from the center UAV. The swarm of UAVs maintains the formation, while performing perception and collaboration tasks at an altitude of 60 meters.

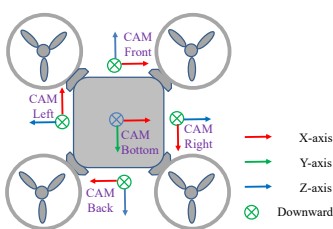

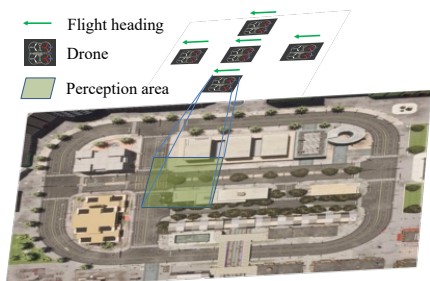

Figure 2: Sensor setup for our data collection platform.

Figure 3: UAV swarm with a cross-shaped formation.

## 3.2 Coordinate Systems

**Global Coordinate System.** The nuScenes dataset uses the East-North-Up (ENU) coordinate system for its global frame. The ENU system is a right-handed Cartesian coordinate system, where the X-axis points to the East, the Y-axis points to the North, and the Z-axis points upward. This system provides a stable, global reference frame for objects and vehicle locations across different scenes in nuScenes. In contrast, the UAV3D dataset, utilizing Unreal Engine 4 (UE4), employs a left-handed Cartesian coordinate system commonly used in real-time rendering and 3D graphics environments. In this system, the X-axis points forward, the Y-axis points to the right, and the Z-axis points upward, following the left-handed rule. Table 2 shows the comparison of the global coordinate systems between nuScenes and UAV3D.

Table 2: Comparison of the global coordinate systems between nuScenes and UAV3D.

|  | nuScenes (ENU) | UAV3D (Unreal Engine 4) |
|---|---|---|
| Type | Right-handed | Left-handed |
| X-axis | Point to East | Point Forward |
| Y-axis | Point to North | Point to Right |
| Z-axis | Point Up | Point Up |
| Application | Autonomous driving, real-world mapping | 3D rendering for games, simulations |

**Ego-UAV Coordinate System.** In UAV3D, the Ego-UAV coordinate system is aligned with the global coordinate system in Unreal Engine 4, and they share the same origin and orientation of the X, Y, and Z axes. This setup simplifies transformations between UAV's local reference frame and the global environment, ensuring consistency in positioning and orientation throughout the dataset.

**Camera Coordinate System.** The camera coordinate system is a 3D coordinate system that describes the position of an object relative to the camera, where the X-axis points to the right of the camera, Y-axis points downwards and Z-axis points forward, away from the camera, representing the depth of an object in the scene. The transformation from world coordinate $\mathbf{P_w}$ to camera coordinate $\mathbf{P_c}$ can be expressed using the following equation:

$$\mathbf{P_c} = \begin{bmatrix} \mathbf{R} & \mathbf{T} \\ \mathbf{0} & 1 \end{bmatrix} \cdot \mathbf{P_w}, \tag{1}$$

where $\mathbf{P_c} = [x_c, y_c, z_c, 1]^T$ is the camera coordinate point, $\mathbf{R}$ is a 3x3 rotation matrix, $\mathbf{T}$ is a 3x1 translation vector, and $\mathbf{P_w} = [x_w, y_w, z_w, 1]^T$ represents the homogeneous coordinates in the world coordinate system. To transform from Unreal Engine 4 (left-handed) to the standard coordinate system (right-handed), the axes need to be rotated and flipped. As a result, a point $(x, y, z)$ in the left-handed system is transformed into the right-handed system as $(x, y, z) \rightarrow (y, -z, x)$. In UAV3D, the transformation from world coordinate $\mathbf{P_w}$ to camera coordinate $\mathbf{P'_c}$ can be expressed as follows:

$$\mathbf{P'_c} = \begin{bmatrix} 0 & 1 & 0 & 0 \\ 0 & 0 & -1 & 0 \\ 1 & 0 & 0 & 0 \\ 0 & 0 & 0 & 1 \end{bmatrix} \cdot \mathbf{P_c} = \begin{bmatrix} 0 & 1 & 0 & 0 \\ 0 & 0 & -1 & 0 \\ 1 & 0 & 0 & 0 \\ 0 & 0 & 0 & 1 \end{bmatrix} \cdot \begin{bmatrix} \mathbf{R} & \mathbf{T} \\ \mathbf{0} & 1 \end{bmatrix} \cdot \mathbf{P_w}, \tag{2}$$

where $\mathbf{P'_c} = [x'_c, y'_c, z'_c, 1]^T$.

**Pixel Coordinates.** Pixel coordinates refer to the 2D coordinates $(u, v)$ on an image plane, representing the location of a point in the image as captured by a camera. The transformation from the 3D camera coordinates $(x'_c, y'_c, z'_c)^T$ to the 2D pixel coordinates $(u, v)$ is typically done using the camera intrinsic matrix. The intrinsic matrix accounts for properties of the camera, such as focal length and the optical center of the image. The transformation is formulated as:

$$\begin{bmatrix} u \\ v \\ 1 \end{bmatrix} = \begin{bmatrix} f_x & 0 & c_x \\ 0 & f_y & c_y \\ 0 & 0 & 1 \end{bmatrix} \cdot \begin{bmatrix} \frac{x'_c}{z'_c} \\ \frac{y'_c}{z'_c} \\ 1 \end{bmatrix}, \tag{3}$$

where $(u, v)$ are the corresponding pixel coordinates, $f_x, f_y$ are the focal lengths of the camera along the $x$ and $y$ axes, and $c_x, c_y$ are center coordinates in the image plane.

### 3.3 Image Data and Annotation

We collect synchronous images from all cameras mounted on the five UAVs, totaling 25 images per sample. To support downstream tasks (e.g., detection, tracking, and semantic segmentation), we extract vehicle annotations from the CARLA simulator, and the camera intrinsics and extrinsics from the AirSim simulator. We offer various annotations, including 3D bounding boxes, and pixel-wise semantic labels. Each 3D bounding box is defined by the location of its center in $x$, $y$, and $z$ coordinates, along with dimensions of width, length, height, and orientation angles (yaw, pitch, roll). UAV3D comprises 500k images and 3.3 million 3D bounding boxes, divided into training, validation, and test splits. There are 17 vehicle categories in the dataset, which is organized in a similar format as the popular nuScenes dataset, with the compatibility to the well-established nuScenes-devkit.

## 4 Experiments

We benchmark four standard perception tasks for UAVs: single-UAV 3D object detection, single-UAV object tracking, collaborative-UAV 3D object detection, and collaborative-UAV object tracking. These tasks are essential for the broad applications of UAVs in various fields. For performance evaluations of the four perception tasks, we utilize the same metrics as those used in the Nuscenes [1] dataset.

**Dataset Format and Split.** Our UAV3D dataset adopts the format of nuScenes. Each scene in our dataset consists of 20 frames. In each scene, 5 UAVs are selected as the collaborating agents, and each frame features data sampled from 5 UAVs. We generate 1,000 scenes, totaling 20,000 frames, from which 700 scenes are used for training, 150 for validation, and 150 for test. Consequently, our dataset comprises 14,000 samples in the training set, 3,000 samples in the validation set, and 3,000 samples in the test set.

**Implementation Details.** The bird's-eye view (BEV), a compact 2D map that accurately describes surrounding objects, is a widely used and effective representation in 3D perception. We therefore employ BEV-based representations across all four tasks. We map all vehicle classes to a single "car" class for the perception tasks. However, the original 17 categories are available for more fine-grained perception tasks. We set the perception range of [-102.4m, 102.4m] for the X-axis and [-102.4m, 102.4m] for the Y-axis, defined in the Ego-UAV Euclidean coordinate system. All baseline models were trained with a batch size of 4 for 24 epochs, with a per-GPU batch size of 1. All experiments were conducted on a computing server equipped with 8 Nvidia A100 GPUs.

**Benchmark Models.** For the single-UAV perception tasks, we consider three recent works in autonomous driving, including PETR [16, 17], BEVFusion [19], and DETR3D [25]. PETR [16] transforms multi-view 2D features into 3D position-aware features by incorporating a 3D positional embedding. The object queries, initialized from 3D space, can directly perceive the 3D object information by interacting with the produced 3D position-aware features. In our implementation of PETR on UAV3D, we experimented with both ResNet-50 and ResNet-101 as backbones. However, PETR's performance with ResNet-101 was inferior to that with ResNet-50. We chose not to include this unexpected result in the paper, as ResNet-101 typically outperforms ResNet-50. Consequently, we adopted only ResNet-50 as the backbone for the PETR baseline.

Table 3: 3D object detection results on the **validation** set of UAV3D. The best results among all the methods are in bold. The symbol "∗" denotes that we have rounded the original value of mAOE from BEVFusion to 1.0.

| Method | Backbone | Size | mAP↑ | NDS↑ | mATE↓ | mASE↓ | mAOE↓ |
|---|---|---|---|---|---|---|---|
| PETR [16] | Res-50 | 704×256 | 0.512 | 0.571 | 0.741 | 0.173 | 0.072 |
| BEVFusion [19] | Res-50 | 704×256 | 0.487 | 0.458 | 0.615 | **0.152** | 1.000∗ |
| DETR3D [25] | Res-50 | 704×256 | 0.430 | 0.509 | 0.791 | 0.187 | 0.100 |
| PETR [16] | Res-50 | 800×450 | 0.581 | 0.632 | 0.625 | 0.160 | **0.064** |
| BEVFusion [19] | Res-101 | 800×450 | 0.536 | 0.582 | 0.521 | 0.154 | 0.343 |
| DETR3D [25] | Res-101 | 800×450 | **0.618** | **0.671** | **0.494** | 0.158 | 0.070 |

BEVFusion [19] is a generic multi-task multi-sensor perception framework, which performs sensor fusion in a shared BEV space and treats foreground, background, geometric and semantic information equally. Since UAV3D contains RGB images without other modality data, we utilize BEVFusion to process the RGB-modality only. DETR3D [25] projects learnable 3D queries in 2D images, and then samples the corresponding features for end-to-end 3D bounding box prediction without NMS post-processing.

For the collaborative-UAV perception tasks, we consider four collaboration algorithms as the benchmark models, including When2Com [14], Who2Com [15], V2VNet [24], and DiscoNet [9], all of which have different communication strategies to transmit and fuse the intermediate features. Lower-bound is the single-UAV perception model without collaboration from other agents. Upper-bound is the collaboration model which transmits the raw image data to other agents. While this approach fully leverages the available perception data, it requires a large amount of transmission bandwidth. As for the implementation of the upper-bound baseline, the 25 images from 5 UAVs are directly utilized by the BEVFusion model.

When2Com [14] employs attention-based mechanism for communication group construction: the agents with high correlation scores would be selected as the collaborators. After the attention-based fusion, the updated features would be fed into model head for perception prediction. Who2Com [15] has a similar communication method with When2Com, but it employs a handshake mechanism to select the agent with the highest attention score. V2VNet [24] utilizes a graph neural network to propagate the information within the agents, and employs a convolutional gated recurrent module to fuse the information from other agents. After several rounds of neural message passing, the updated features are fed into the model head to generate perception results. DiscoNet [9] utilizes a directed collaboration graph with matrix-valued edge weight to adaptively choose the informative spatial regions and reject the noisy regions of image features sent by other agents. After the message fusion, the updated features would be transmitted to the model head for perception results.

## 4.1 Single-UAV 3D Object Detection

**Problem Definition.** As one of the most critical perception tasks for UAVs, 3D object detection is designed to localize and recognize objects in 3D space using a single frame. Subsequent tracking tasks depend heavily on the accuracy of these detection results. The models for this task process images from five different perspectives and generate predictions of 3D bounding boxes.

**Backbone and Evaluation.** For the three benchmark models, namely PETR, BEVFusion, and DETR3D, we utilize two backbone architectures, ResNet-50 and ResNet-101, with two different image sizes: (704, 256) and (800, 450). We apply the BEV detection evaluation metrics as used in nuScenes, including NDS, mAP, mATE, mASE, and mAOE. The definitions of these metrics are the same as in Nuscenes [1]. Compared to nuScenes, we do not include the mAVE and mAAE metrics. Therefore, NDS is defined as follows:

$$\text{NDS} = \frac{1}{8}[5\,\text{mAP} + \sum_{\text{mTP} \in \mathbb{TP}} (1 - \min(1,\,\text{mTP}))], \tag{4}$$

where $\mathbb{TP}$ is a set of the three mean True Positive metrics: mATE, mASE, and mAOE. We focus on analyzing the detection results and report them for both validation and test sets.

**Quantitative Results.** Table 3 reports the results of 3D object detection. Among the three baselines, DETR3D achieves the best detection results in three of five metrics, including mAP, NDS and mATE.

Table 4: 3D object detection results on the **test** set of UAV3D. The best results among all the methods are in bold. The symbol "∗" has the same meanings as in Table 3.

| Method | Backbone | Size | mAP↑ | NDS↑ | mATE↓ | mASE↓ | mAOE↓ |
|---|---|---|---|---|---|---|---|
| PETR [16] | Res-50 | 704×256 | 0.507 | 0.568 | 0.748 | 0.173 | 0.069 |
| BEVFusion [19] | Res-50 | 704×256 | 0.485 | 0.456 | 0.619 | 0.153 | 1.000* |
| DETR3D [25] | Res-50 | 704×256 | 0.424 | 0.505 | 0.794 | 0.187 | 0.094 |
| PETR [16] | Res-50 | 800×450 | 0.575 | 0.627 | 0.632 | 0.159 | **0.061** |
| BEVFusion [19] | Res-101 | 800×450 | 0.536 | 0.583 | 0.518 | **0.156** | 0.339 |
| DETR3D [25] | Res-101 | 800×450 | **0.610** | **0.665** | **0.501** | 0.157 | 0.068 |

Table 5: Object tracking results on the **validation** set of UAV3D. The best results among all the methods are in bold.

| Method | Backbone | Size | AMOTA↑ (%) | AMOTP↓ (m) | MOTA↑ (%) | MOTP↓ (m) | TID↓ (s) | LGD↓ (s) |
|---|---|---|---|---|---|---|---|---|
| PETR [16] | Res-50 | 704×256 | 0.199 | 1.294 | 0.195 | 0.794 | 1.280 | 2.970 |
| BEVFusion [19] | Res-50 | 704×256 | 0.566 | 1.137 | 0.501 | 0.695 | 0.790 | 1.600 |
| DETR3D [25] | Res-50 | 704×256 | 0.089 | 1.382 | 0.121 | 0.800 | 1.540 | 3.530 |
| PETR [16] | Res-50 | 800×450 | 0.291 | 1.156 | 0.256 | 0.677 | 1.090 | 2.550 |
| BEVFusion [19] | Res-101 | 800×450 | **0.606** | **1.006** | **0.540** | 0.627 | **0.700** | **1.390** |
| DETR3D [25] | Res-101 | 800×450 | 0.262 | 1.123 | 0.238 | **0.561** | 1.140 | 2.720 |

The results of the three baselines are comparable in terms of mASE, although BEVFusion performs slightly better than the other two baselines. In terms of mAOE, PETR and DETR3D significantly outperform BEVFusion. On the other hand, using larger backbones and input sizes can significantly enhance baseline performance. For instance, DETR3D shows substantial improvements, with gains of approximately 0.18 in mAP and 0.16 in NDS. Meanwhile, BEVFusion has the modest increases, with gains of about 0.05 in mAP and 0.13 in NDS. Table 4 presents the results of 3D object detection on the test set, which are consistent with those observed on the validation set.

## 4.2 Single-UAV Object Tracking

**Problem Definition.** Unlike the object detection, the object tracking requires the production of temporally consistent perception results. In this task, object tracking involves utilizing bounding boxes and object identities to track various objects across a temporal sequence.

**Evaluation Metrics.** We employ the object tracking evaluation metrics defined in nuScenes [1], including Average Multi Object Tracking Accuracy (AMOTA), Average Multi Object Tracking Precision (AMOTP), Track Initialization Duration (TID), and Longest Gap Duration (LGD). In addition, the traditional metrics are also employed, such as Object Tracking Accuracy (MOTA) and Object Tracking Precision (MOTP). While MOTA can measure detection errors and association errors, MOTP measures localization accuracy.

**Quantitative Results.** Table 5 reports the results of object tracking on the validation set. BEVFusion achieves the best tracking performance across five of six metrics, specifically AMOTA, AMOTP, MOTA, TID, and LGD, and achieves the second-best result in MOTP. The baselines PETR and DETR3D demonstrate comparable performances, yet they lag behind BEVFusion by a large margin, with a difference of approximately 0.3 in AMOTA. Similar to the single-UAV object detection task, the object tracking performance of the three baselines benefits from the use of larger backbones and input image sizes. DETR3D achieves significant improvements, with gains of approximately 0.18 in AMOTA and 0.11 in MOTA. In contrast, BEVFusion shows smaller increases, with about 0.04 in AMOTA and 0.04 in MOTA. Table 6 shows the results of object tracking on test set, which are consistent with those observed on the validation set.

## 4.3 Collaborative-UAV 3D Object Detection

**Problem Definition.** Similar to single-UAV 3D object detection, the objective of collaborative-UAV 3D object detection is to localize and recognize objects in 3D space using a single frame. However, in collaborative-UAV 3D object detection, one agent can collaborate with other agents to enhance perception performance.

Table 6: Single-UAV object tracking results on the **test** set of UAV3D. The best results among all the methods are in bold.

| Method | Backbone | Size | AMOTA↑ | AMOTP↓ | MOTA↑ | MOTP↓ | TID↓ | LGD↓ |
|---|---|---|---|---|---|---|---|---|
| | | | (%) | (m) | (%) | (m) | (s) | (s) |
| PETR [16] | Res-50 | 704×256 | 0.198 | 1.295 | 0.193 | 0.792 | 1.300 | 2.940 |
| BEVFusion [19] | Res-50 | 704×256 | 0.561 | 1.140 | 0.494 | 0.701 | 0.790 | 1.580 |
| DETR3D [25] | Res-50 | 704×256 | 0.091 | 1.382 | 0.122 | 0.801 | 1.590 | 3.610 |
| PETR [16] | Res-50 | 800×450 | 0.288 | 1.157 | 0.252 | 0.681 | 1.13 | 2.61 |
| BEVFusion [19] | Res-101 | 800×450 | **0.600** | **1.039** | **0.533** | 0.615 | **0.710** | **1.440** |
| DETR3D [25] | Res-101 | 800×450 | 0.259 | 1.125 | 0.238 | **0.563** | 1.190 | 2.730 |

Table 7: Collaborative-UAV 3D object detection results on the **validation** set of UAV3D. The best results among all the methods are in bold.

| Method | mAP↑ | NDS ↑ | mATE↓ | mASE↓ | mAOE↓ |
|---|---|---|---|---|---|
| Lower-bound | 0.544 | 0.556 | 0.540 | 0.147 | 0.578 |
| When2Com [14] | 0.550 | 0.507 | 0.534 | 0.156 | 0.679 |
| Who2Com [15] | 0.546 | 0.597 | 0.541 | 0.150 | 0.263 |
| V2VNet [24] | 0.647 | 0.628 | 0.508 | 0.167 | 0.533 |
| DiscoNet [9] | 0.700 | 0.689 | 0.423 | 0.143 | 0.422 |
| Upper-bound | **0.720** | **0.748** | **0.391** | **0.106** | **0.117** |

Table 8: Collaborative-UAV 3D object detection results on the **test** set of UAV3D. The best results among all the methods are in bold.

| Method | mAP↑ | NDS ↑ | mATE↓ | mASE↓ | mAOE↓ |
|---|---|---|---|---|---|
| Lower-bound | 0.542 | 0.558 | 0.543 | 0.148 | 0.549 |
| When2Com [14] | 0.547 | 0.505 | 0.539 | 0.156 | 0.659 |
| Who2Com [15] | 0.541 | 0.594 | 0.548 | 0.150 | 0.255 |
| V2VNet [24] | 0.645 | 0.626 | 0.517 | 0.166 | 0.527 |
| DiscoNet [9] | 0.696 | 0.686 | 0.428 | 0.143 | 0.417 |
| Upper-bound | **0.720** | **0.746** | **0.398** | **0.107** | **0.121** |

**Model and Evaluation.** We employ the BEVFusion with backbone Swin-transformer [18] as the basic framework, and incorporate different communication strategies to the benchmark, including Lower-bound, When2Com [14], Who2Com [15], V2VNet [24], DiscoNet [9], and Upper-bound. We employ the 3D object detection metrics as in nuScenes: NDS, mAP, mATE, mASE, and mAOE. We target the detection results and report them on the validation and test sets.

**Quantitative Results.** Table 7 reports the results of collaborative-UAV 3D object detection on validation set. The Upper-bound baseline has achieved the best performance and outperformed the Lower-bound baseline by a large gap of 0.18 in mAP and 0.19 in NDS because the Upper-bound baseline can fully utilize the perception image data from all the five collaborative UAVs. DiscoNet has achieved the best performance in the four collaborative baselines, with a slight performance gap of 0.02 in mAP and 0.06 in NDS, compared to the Upper-bound. V2VNet has the second best performance due to the effective fusion of image features. The When2Com and Who2Com baselines have slightly better performance than the Lower-bound baseline since the fusion mechanism can only choose parts of intermediate image features from other agents. Specifically, When2Com has a threshold score to filter out the image features with low attention score, while Who2Com chooses the image features with the larges attention score from one of the collaborative agents. Table 8 reports the results of collaborative-UAV 3D object detection on test set, which again are consistent with those observed on the validation set.

## 4.4 Collaborative-UAV Object Tracking

**Problem Definition.** Similar to single-UAV object tracking, the objective of collaborative-UAV object tracking is to use bounding boxes, and object identities to track different objects within a temporal sequence. However, in collaborative-UAV 3D object tracking, one agent can collaborate with other agents to boost the tracking performance.

Table 9: Collaborative-UAV object tracking results on the **validation** set of UAV3D. The best results among all the methods are in bold.

| Method | AMOTA↑ | AMOTP↓ | MOTA↑ | MOTP↓ | TID↓ | LGD↓ |
|--------|--------|--------|-------|-------|------|------|
| | (%) | (m) | (%) | (m) | (s) | (s) |
| Lower-bound | 0.644 | 1.018 | 0.593 | 0.611 | 0.620 | 1.280 |
| When2Com [14] | 0.646 | 1.012 | 0.595 | 0.618 | 0.590 | 1.200 |
| Who2Com [15] | 0.648 | 1.012 | 0.602 | 0.623 | 0.580 | 1.200 |
| V2VNet [24] | 0.782 | 0.803 | 0.735 | 0.587 | 0.360 | 0.710 |
| DiscoNet [9] | 0.809 | 0.703 | 0.766 | 0.516 | 0.300 | 0.590 |
| Upper-bound | **0.812** | **0.672** | **0.781** | **0.476** | **0.300** | **0.570** |

Table 10: Collaborative-UAV object tracking results on the **test** set of UAV3D. The best results among all the methods are in bold.

| Method | AMOTA↑ | AMOTP↓ | MOTA↑ | MOTP↓ | TID↓ | LGD↓ |
|--------|--------|--------|-------|-------|------|------|
| | (%) | (m) | (%) | (m) | (s) | (s) |
| Lower-bound | 0.641 | 1.020 | 0.590 | 0.614 | 0.610 | 1.270 |
| When2Com [14] | 0.643 | 1.016 | 0.591 | 0.611 | 0.610 | 1.250 |
| Who2Com [15] | 0.645 | 1.017 | 0.597 | 0.631 | 0.600 | 1.220 |
| V2VNet [24] | 0.778 | 0.811 | 0.727 | 0.598 | 0.360 | 0.710 |
| DiscoNet [9] | 0.807 | 0.707 | 0.760 | 0.512 | 0.330 | 0.650 |
| Upper-bound | **0.810** | **0.679** | **0.772** | **0.485** | **0.300** | **0.580** |

**Model and Evaluation.** Following the collaborative-UAV 3D object detection, we employ the BEVFusion with backbone Swin-transformer [18] as the basic framework, and incorporate different communication strategies to the benchmarks, including Lower-bound, When2Com [14], Who2Com [15], V2VNet [24], DiscoNet [9], and Upper-bound. Regarding the evaluation metrics, we employ AMOTA, AMOTP, MOTA, MOTP, TID, and LGD as those used in the single-UAV object tracking task.

**Quantitative Results.** Table 9 reports the results of collaborative-UAV object tracking on the validation set. The Upper-bound baseline has shown a large performance gain against the Lower-bound baseline by about 0.17 in AMOTA and 0.44 in AMOTP, which demonstrates the effectiveness of collaborative perception for utilizing image data from other UAVs. DiscoNet has achieved the best performance in the four intermediate feature fusion baselines, with a large performance improvement of 0.16 in AMOTA and 0.31 in AMOTP, compared to the Low-bound. V2VNet has the second-best performance among the four baselines due to the effective fusion of image features. The When2Com and Who2Com baselines have shown better performance than Lower-bound due to the attention mechanism for fusing intermediate image features from other agents. Table 10 shows the results of collaborative-UAV object tracking on the test set, which are consistent with those observed on the validation set.

## 4.5 Visualization

Figure 4 presents the qualitative results of 3D object detection of the six collaborative baselines in the challenging urban environment of Town 10. As shown in Figure 4(a), the Lower-bound model exhibits relatively lower detection performance in terms of bounding box size and orientation, and it fails to predict objects obstructed by tall buildings due to the occlusions. Figure 4(f) illustrates that the predicted 3D bounding boxes (in red) generated by the Upper-bound baseline closely align with the ground truth annotations (in blue) in terms of box size and orientation. Notably, the Upper-bound model successfully detects vehicles even when they are obscured by tall buildings. This illustrates the effectiveness of collaborative perception in overcoming occlusions in complex urban environments. Regarding some failure cases, certain missing bounding boxes fall outside the detection range specified in our settings. Additionally, there remains substantial room of improvement for advanced model designs. The four intermediate feature fusion methods outperform the Lower-bound baseline, demonstrating the capability to detect objects obscured by tall buildings. Among the four baselines, When2Com and Who2Com exhibit relatively lower performance, while DiscoNet has demonstrated the best performance, with the detection results closely aligned with those of the Upper-bound baseline. Visualizations of the front, bottom, left, and right views can be found in the Appendix.

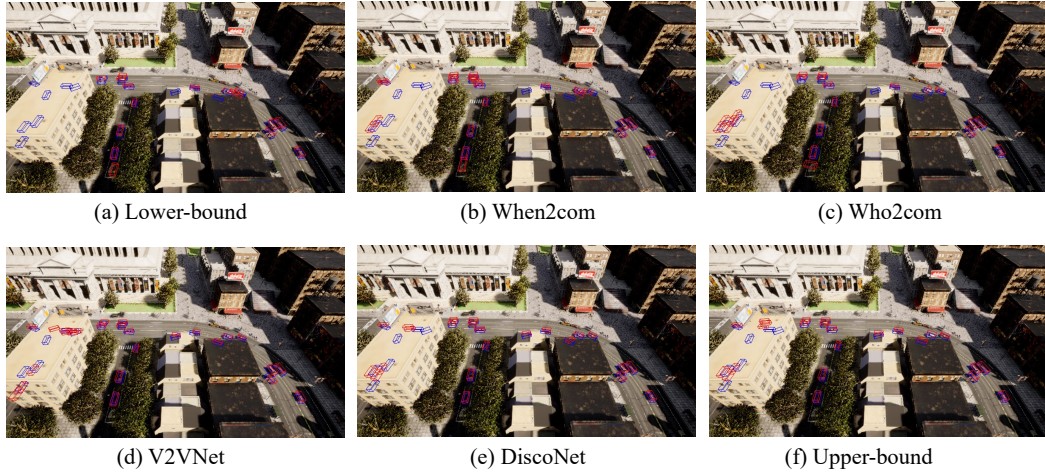

| (a) Lower-bound | (b) When2com | (c) Who2com |
| (d) V2VNet | (e) DiscoNet | (f) Upper-bound |

Figure 4: Visualization of the **back view** for collaborative-UAV 3D object detection results on UAV3D. Red boxes represent the predictions, while blue boxes indicate the ground truth.

## 5  Limitations

Real-world datasets typically involve more complex conditions, including localization errors, sensor synchronization issues, and communication latency between agents. In contrast, simulated datasets benefit from the controlled, ideal settings of a simulation environment. Apparently, there is a domain gap between the real and simulated datasets, where the performance would degrade when applying models trained with simulated data to the real-world applications. The typical approach to mitigate the domain gap is to the domain adaptation technique, which trains the model on the simulated dataset and adapt the model to real-world applications. To the best of our knowledge, the real-world 3D dataset for UAVs is not publicly available due to security, privacy concerns, and the expensive cost of data annotation. We can adapt the sim-to-real domain adaption algorithms in autonomous driving [27, 7] to the UAV3D dataset. Moreover, to evaluate perception tasks on UAV3D, we have selected three well-known baseline models from the autonomous driving community: BEVFusion, DETR3D, and PETR. Recently, several advanced models for 3D object detection in autonomous driving, such as StreamPETR [23], Sparse4DV2 [12], BEVNeXt [10], and SparseBEV [13], have been proposed. These models can be adapted to UAV3D and potentially can achieve even better performance. We leave this exploration for future work.

## 6  Conclusion

We introduce UAV3D – a large-scale 3D perception benchmark for Unmanned Aerial Vehicles – collected from the CARLA-AirSim co-simulation environment. UAV3D features the largest collection of RGB images and 3D bounding-box annotations compared to previously released UAV datasets, and supports a wide range of 3D perception tasks. To facilitate UAV-related research, we have benchmarked several recent perception models from autonomous driving, adapting them to single-UAV object detection and tracking, and collaborative-UAV object detection and tracking. The whole benchmark, including the UAV3D dataset and source code, is made publicly available, with the aim to facilitate research in UAV-based 3D perception. Future work includes incorporating a diverse range of times (day and night) and weather conditions (sun, rain, and clouds) in the data collection, simulating latency issues, and exploring more advanced models for 3D perception. We believe our work will inspire many relevant research including but not limited to multi-agent reinforcement learning and collaborative learning systems.

## Acknowledgments and Disclosure of Funding

The software and data were created by Georgia State University Research Foundation under Army Research Laboratory (ARL) Award Numbers W911NF-22-2-0025 and W911NF-23-2-0224. ARL, as the Federal awarding agency, reserves a royalty-free, nonexclusive and irrevocable right to reproduce, publish, or otherwise use this software for Federal purposes, and to authorize others to do so in accordance with 2 CFR 200.315(b).

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

# A  Appendix

## A.1  Annotation Statistics

We provide more details on the annotations of UAV3D here. There are 17 vehicle categories in UAV3D. As shown in Figure 5, the categories are almost evenly distributed as they were generated randomly from the simulation environment. In our benchmark experiments, we treated all vehicles as a single "car" class for the perception task. The 17 categories are available for more fine-grained perception tasks.

## A.2  UAV3D vs. CoPerception-UAVs

As shown in Table 11, UAV3D and CoPerception-UAVs are both UAV datasets that utilize simulations in Carla and AirSim, but they differ in several aspects. UAV3D includes more maps (Towns 3, 6, 7, and 10) compared to CoPerception-UAVs (Towns 3, 4, and 6) and operates with a static cross-shaped drone formation, while CoPerception-UAVs adopts a static V-shaped formation and a dynamic shape. Additionally, UAV3D provides 3D boxes, semantic labels, and box velocity as labels, whereas CoPerception-UAVs focus only on 3D boxes and semantic labels. UAV3D also significantly surpasses CoPerception-UAVs in terms of sample size, with 20,000 samples and 500,000 images, compared to 5,276 samples and 131,900 images in CoPerception-UAVs. Finally, UAV3D offers public access to its source code, while CoPerception-UAVs does not as of the release of the UAV3D benchmark.

Table 11: Comparison between CoPerception-UAVs and UAV3D.

|  | CoPerception-UAVs | UAV3D |
| --- | --- | --- |
| Simulation | Carla & AirSim | Carla & AirSim |
| Maps | Town 3, 4 &6 | Town 3, 6, 7&10 |
| Agents | 5 drones | 5 drones |
| Sensors | 25 RGB cameras | 25 RGB cameras |
| Height | 40, 60, 80m | 60m |
| Formation | Static V shape & Dynamic shape | Static cross-shape |
| #Categories | 21 | 17 |
| Frequency | 0.25Hz | 0.5Hz |
| Labels | 3D boxes & Semantic labels | 3D boxes & Semantic labels & Box velocity |
| #Samples | 5,276 | 20,000 |
| #Images | 131,900 | 500,000 |
| source code | ✗ | ✓ |

## A.3  Additional Experimental Results

For collaborative-UAV 3D object detection, we employ the additional metrics of Average Precision (AP) at Intersection-over-Union (IoU) thresholds of 0.5 and 0.7, following V2X-Sim [8]. Note that the orientation of boxes is not considered in our AP calculations. As shown in Table 12, the Upper-bound baseline has achieved the best performance and outperformed the Lower-bound baseline by a large gap of 0.213 in $AP@0.5$ and 0.183 in $AP@0.7$. DiscoNet has achieved the best performance in the four collaborative baselines, with a slight performance gap of 0.087 in $AP@0.5$ and 0.078 in $AP@0.7$, compared to the Upper-bound. The When2Com and Who2Com baselines have slightly better performance than the Lower-bound baseline since the fusion mechanism can only choose parts intermediate image features from other agents. The results of collaborative-UAV 3D object detection on test set are consistent with those observed on the validation set.

## A.4  UAV3D V1.0-mini

To facilitate the usage of UAV3D for perception tasks, a mini version of UAV3D (UAV3D V1.0-mini) has been created. UAV3D V1.0-mini is significantly smaller in size at just 7 GB, compared to the full UAV3D V1.0, which is approximately 400 GB. Despite the reduced size, UAV3D V1.0-mini includes 20 scenes captured from Town 10, with 400 samples and 10,000 images. In contrast, the full version, UAV3D V1.0, includes 1,000 scenes from multiple towns (Towns 3, 6, 7, and 10), providing 20,000 samples and 500,000 images. This mini version enables users to work with UAV3D more easily, particularly in environments with limited computational resources. The comparison between the UAV3D and UAV3D v1.0-mini is shown in Table 13.

Table 12: Collaborative-UAV 3D object detection results on the **validation** and **test** sets of UAV3D. The best results among all the methods are in bold.

| Method | Validation set | | Test set | |
| --- | --- | --- | --- | --- |
| | AP@IoU=0.5 ↑ | AP@IoU=0.7 ↑ | AP@IoU=0.5 ↑ | AP@IoU=0.7 ↑ |
| Lower-bound | 0.454 | 0.128 | 0.457 | 0.140 |
| When2Com [14] | 0.456 | 0.164 | 0.461 | 0.166 |
| Who2Com [15] | 0.447 | 0.161 | 0.453 | 0.141 |
| V2VNet [24] | 0.537 | 0.169 | 0.545 | 0.141 |
| DiscoNet [9] | 0.580 | 0.233 | 0.649 | 0.247 |
| Upper-bound | **0.667** | **0.311** | **0.673** | **0.316** |

Table 13: Comparison between UAV3D V1.0-mini and UAV3D.

| | UAV3D V1.0-mini | UAV3D |
| --- | --- | --- |
| Maps | Town 10 | Town 3, 6, 7&10 |
| #scenes | 20 | 1,000 |
| #Samples | 400 | 20,000 |
| #Images | 10,000 | 500,000 |
| size | 7GB | 400 GB |

Table 14: Single-UAV 3D object detection results on the validation set of UAV3D V1.0-mini.

| Method | Dataset | Backbone | Size | mAP↑ | NDS↑ | mATE↓ | mASE↓ | mAOE↓ |
| --- | --- | --- | --- | --- | --- | --- | --- | --- |
| PETR | V1.0-mini | Res-50 | 704×256 | 0.003 | 0.101 | 1.000 | 0.200 | 1.000 |
| BEVFusion | V1.0-mini | Res-50 | 704×256 | 0.113 | 0.280 | 0.912 | 0.198 | 0.212 |
| DETR3D | V1.0-mini | Res-50 | 704×256 | 0.023 | 0.114 | 1.000 | 0.197 | 1.000 |
| PETR | V1.0-mini | Res-50 | 800×450 | 0.005 | 0.103 | 1.000 | 0.198 | 1.000 |
| BEVFusion | V1.0-mini | Res-101 | 800×450 | 0.112 | 0.273 | 0.947 | 0.195 | 0.232 |
| DETR3D | V1.0-mini | Res-101 | 800×450 | 0.051 | 0.174 | 1.000 | 0.195 | 0.663 |

Table 15: Collaborative-UAV 3D object detection results on the validation set of UAV3D V1.0-mini.

| Method | Dataset | mAP↑ | NDS ↑ | mATE↓ | mASE↓ | mAOE↓ |
| --- | --- | --- | --- | --- | --- | --- |
| Lower-bound | V1.0-mini | 0.147 | 0.304 | 0.933 | 0.192 | 0.176 |
| When2Com | V1.0-mini | 0.151 | 0.310 | 0.922 | 0.193 | 0.156 |
| Who2Com | V1.0-mini | 0.153 | 0.314 | 0.899 | 0.193 | 0.161 |
| V2VNet | V1.0-mini | 0.265 | 0.390 | 0.855 | 0.192 | 0.152 |
| DiscoNet | V1.0-mini | 0.218 | 0.322 | 0.886 | 0.193 | 0.431 |
| Upper-bound | V1.0-mini | 0.349 | 0.468 | 0.656 | 0.190 | 0.153 |

**Experimental Results** Table 14 reports the results of single-UAV 3D object detection on the validation set of UAV3D V1.0-mini. Among these models, BEVFusion consistently achieves the best performance across all five metrics, including mAP, NDS, mATE, mASE, and mAOE. PETR, on the other hand, shows the worst overall performance and is particularly sensitive to the scale of the training dataset. When larger backbone models (ResNet-101) and larger input image sizes (800×450) are used, DETR3D demonstrates modest improvement in its detection performance, while PETR and BEVFusion show almost no improvement.

Table 15 shows the collaborative-UAV 3D object detection results on the validation set of UAV3D V1.0-mini. Among the four collaborative baselines, V2VNet achieves the best performance with mAP of 0.265 and NDS of 0.390, outperforming When2Com, Who2Com, and DiscoNet. DiscoNet performs relatively well, showing higher performance in mAP and NDS compared to When2Com and Who2Com, though it suffers from a high mAOE of 0.431. The lower-bound baseline has the worst performance, while the upper-bound baseline provides the best performance with mAP of 0.349 and NDS of 0.468.

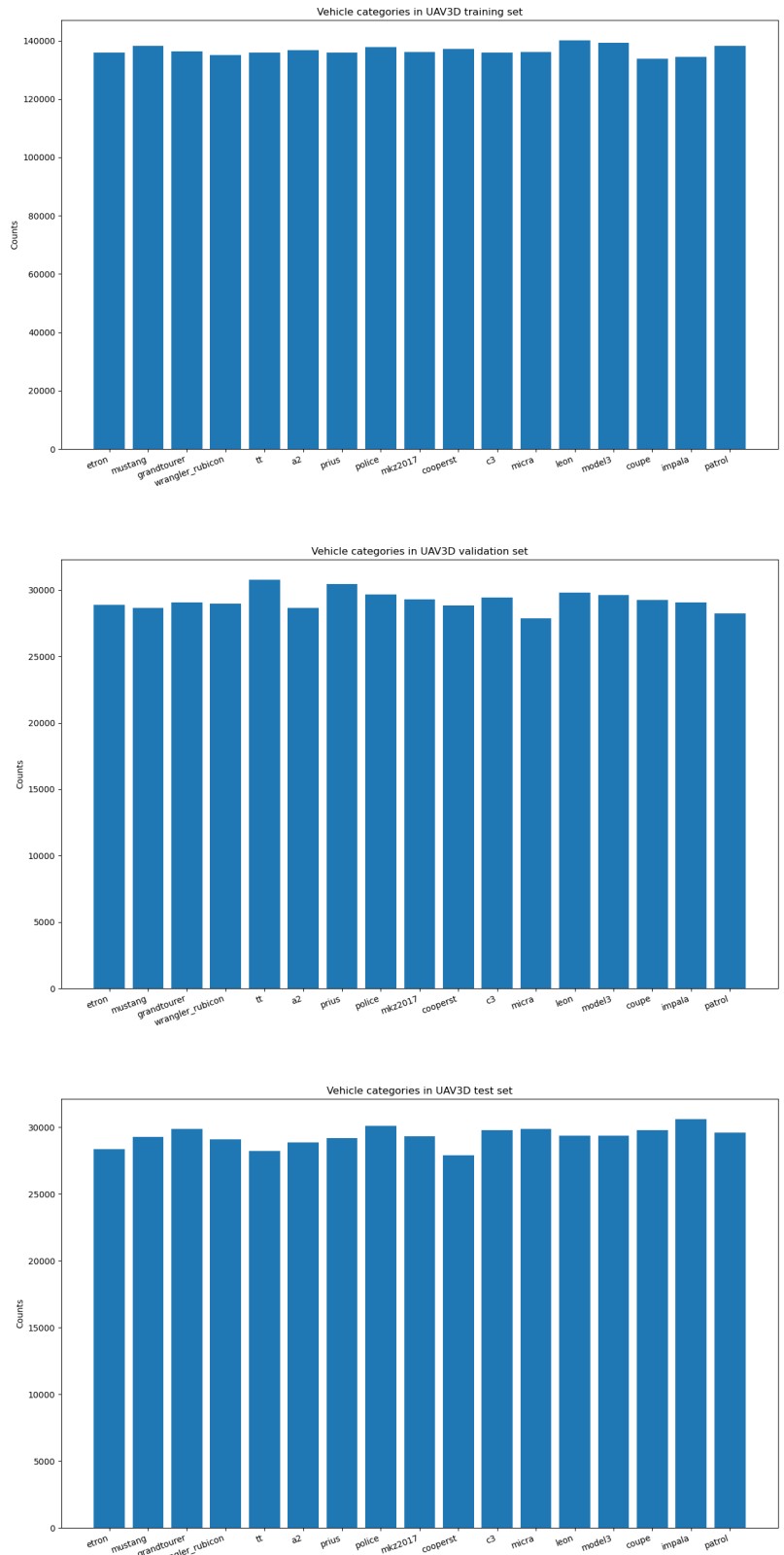

Figure 5: Number of annotations per category in UAV3D training, validation and test set. The vehicle categories are almost evenly distributed.

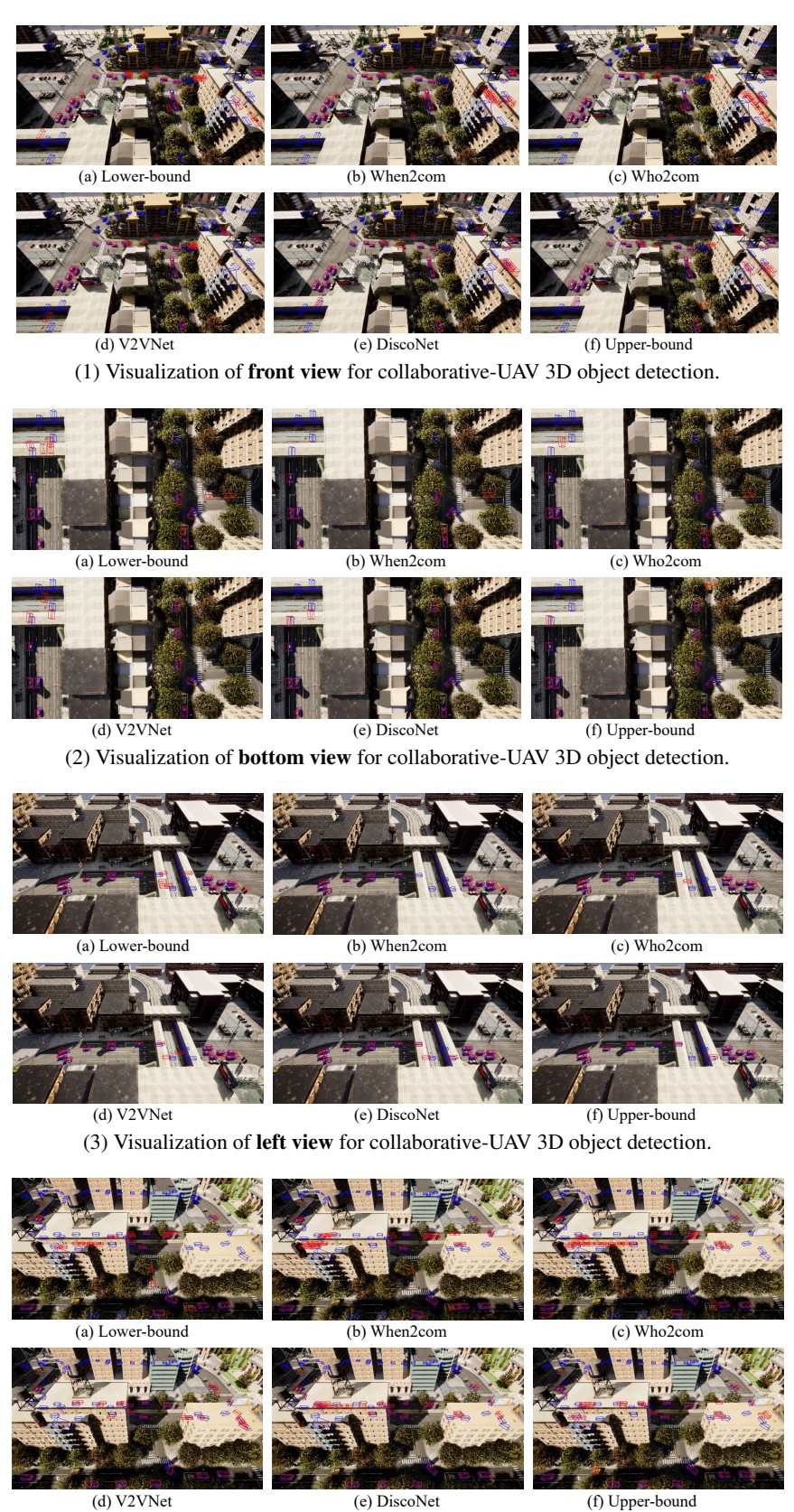

(a) Lower-bound     (b) When2com     (c) Who2com

(d) V2VNet     (e) DiscoNet     (f) Upper-bound

(1) Visualization of **front view** for collaborative-UAV 3D object detection.

(a) Lower-bound     (b) When2com     (c) Who2com

(d) V2VNet     (e) DiscoNet     (f) Upper-bound

(2) Visualization of **bottom view** for collaborative-UAV 3D object detection.

(a) Lower-bound     (b) When2com     (c) Who2com

(d) V2VNet     (e) DiscoNet     (f) Upper-bound

(3) Visualization of **left view** for collaborative-UAV 3D object detection.

(a) Lower-bound     (b) When2com     (c) Who2com

(d) V2VNet     (e) DiscoNet     (f) Upper-bound

(4) Visualization of **right view** for collaborative-UAV 3D object detection.

Figure 6: Visualizations of collaborative-UAV 3D object detection results on UAV3D. Red boxes represent the predictions, while blue boxes indicate the ground truth.

