# OpenReview forum: "UAV3D: A Large-scale 3D Perception Benchmark for Unmanned Aerial Vehicles"
_NeurIPS.cc/2024/Datasets_and_Benchmarks_Track — NeurIPS 2024 Track Datasets and Benchmarks Poster_

### Official Review · Reviewer_xNiq · 2024-07-24
**A large-scale synthesized dataset for UAV 3D perception.**

**Rating:** 6
**Confidence:** 3
**Correctness:** Yes, their claims are correct.
**Clarity:** Yes, this paper is well written and e…

**Review:**

Their main contribution is the development of the UAV3D dataset, which is the largest collection of RGB images and 3D box annotations compared to existing UAV datasets. This dataset supports four representative perception tasks, including both single-UAV perception and multi-UAV collaboration. They utilize a CARLA and AirSim co-simulation to collect the dataset, providing a flexible and extensible environment for implementing more interactive tasks.

However, despite the dataset's size, they did not demonstrate how much it can improve existing 3D perception methods. Additionally, since all data are collected in simulation, it is crucial to evaluate how representative the data is of real-world conditions and discuss how models trained on this dataset can be transferred to real-world applications.

**Strengths:**

This paper provide a large-scale 3D perception dataset for UAVs and establish a CARLA and AirSim co-simulation for data collection.

**Additional Feedback:**

All comments have been mentioned before.

**Documentation:**

Yes, they provide sufficient detail on data collection and organization, and they provide a link to access their code and dataset.

**Limitations:**

No, they didn't adequately addressed the limitations.

**Opportunities For Improvement:**

I would like to see the following points addressed:

1. **Sim-to-Real Transfer**: Please include a discussion on the sim-to-real transfer process. An experimental analysis highlighting the effectiveness and challenges of transferring simulations to real-world applications would be appreciated.
2. **Interactive Tasks in Co-Simulation**: It would be beneficial to demonstrate interactive tasks within a co-simulation framework. Specifically, showcasing how a multi-UAV system can collaborate and perceive interactively in a large-scale environment would provide valuable insights.
3. **Evidence of Improvement in 3D Perception**: As I am not a professional in 3D perception, it would be helpful to see evidence of how much the existing methods trained on UAV3D have improved. Comparisons with current datasets and methods would provide a clearer understanding of its advantages.

**Relation To Prior Work:**

Yes, they clearly compare their dataset with existing ones.

**Summary And Contributions:**

They introduce a new dataset for single-agent and collaborative 3D perception in UAVs, synthesized using the CARLA environment with AirSim for UAV simulation. The dataset, UAV3D, includes 1,000 scenes, each containing 20 frames fully annotated with 3D bounding boxes for vehicles. They conduct benchmarks for four 3D perception tasks: single UAV 3D object detection, single UAV object tracking, collaborative UAVs 3D object detection, and collaborative UAVs object tracking. Additionally, established methods have been applied to these tasks to provide baseline results for this benchmark.

---

> ### Author Rebuttal · Authors · 2024-08-16
>
> Our project website has been transferred to GitHub: https://huiyegit.github.io/UAV3D_Benchmark/.
>
> **Opportunities For Improvement:**
>
> 1. Sim-to-Real Transfer: Please include a discussion on the sim-to-real transfer process. An experimental analysis highlighting the effectiveness and challenges of transferring simulations to real world applications would be appreciated.
>
> Thank you for this suggestion. Real-world datasets typically involve more complex conditions, including localization errors, sensor synchronization issues, and communication latency between agents. In contrast, simulated datasets benefit from the controlled, ideal settings of a simulation environment.
> Apparently, there is a domain gap between the real and simulated datasets, where the performance would degrade when applying  models trained with simulated data to the real-world applications. The typical approach to mitigate the domain gap is to apply the Domain Adaptation, which trains the model on the simulated dataset and adapt the model to real-world applicaitons.
>
> To the best of our knowledge, the real 3D dataset for UAVs is not publicly available, due to security and privacy concerns, as well as the expensive cost of data labeling. We can refer to the sim-to-real transfer problem in autonomous driving. For example, V2V4Real [1] presents the Sim2Real Domain Adaptation to reduce domain discrepancy in the cooperative 3D detection task. S2R-ViT[2] presents a model architecture to bridge the gap from simulation to reality for Multi-Agent Cooperative Perception.
> We will add the discussion of the limitations in the paper.
>
> [1] Xu, Runsheng, et al. "V2v4real: A real-world large-scale dataset for vehicle-to-vehicle cooperative perception." Proceedings of the IEEE/CVF Conference on Computer Vision and Pattern Recognition. 2023.
>
> [2] Li, Jinlong, et al. "S2r-vit for multi-agent cooperative perception: Bridging the gap from simulation to reality." 2024 IEEE International Conference on Robotics and Automation (ICRA). IEEE, 2024.
>
> 2. Interactive Tasks in Co-Simulation: It would be beneficial to demonstrate interactive tasks within a co-simulation framework. Specifically, showcasing how a multi-UAV system can collaborate and perceive interactively in a large-scale environment would provide valuable insights.
>
> Thank you for the suggestion. We will create a video to illustrate how multiple UAVs collaborate and perceive step-by-step in a large scale environment: when some vehicles are occluded by tall buildings from ego-UAV but only visible by 1-2 collaborator UAVs. We will add this video in the project website.
>
> 3. Evidence of Improvement in 3D Perception: As I am not a professional in 3D perception, it would be helpful to see evidence of how much the existing methods trained on UAV3D have improved. Comparisons with current datasets and methods would provide a clearer understanding of its advantages.
>
> Thank you for the suggestion. The comparison between different datasets requires the same model and experimental settings. However, there is currently no publicly available **source code** to utilize CoPerception-UAVs. The official repo of CoPerception-UAVs does not support this dataset. Please refer to the official repo: https://github.com/MediaBrain-SJTU/Where2comm.
>
> The CoPerception-UAVs has 183 scenes and 131.9k images. We therefore created a similar sized dataset UAV3D-200, with 200 scenes and 100k images from UAV3D to mimic CoPerception-UAVs. As shown in Table below, the performances of  three baseline models with all 15 metrics on UAV3D are better  than those on UAV3D-200, demonstrating that scaling up the UAV dataset improves the performance. We will include this result in the appendix.
>
>
>     Table: Comparison of 3D object detection results between UAV3D-200 and UAV3D. UAV3D-200 and UAV3D have 200 and 1,000 scenes, respectively.
> |  Model | Dataset | Backbone | Size  | ....mAP↑....  |.... NDS↑....  | ....mATE↓....  | ....mASE↓....  | ....mAOE↓....  |
> | :--: | :-------: | :--: | :--: | :--: | :--: | :--: | :--: | :--: |
> | PETR  |  UAV3D-200 |   Res-50 |  800*450   |  0.338 |  0.424 | 1.000 |  0.189 |  0.109 |
> | PETR  |  UAV3D        |   Res-50 |  800*450    | 0.581(+0.243)  | 0.632 (+0.208)   |  0.625 (+0.375)  |   0.160 (+0.029)}   |  0.064(+0.045)}  |
> | BEVFusion |  UAV3D-200  |  Res-101  | 800*450    |  0.511 | 0.492  | 0.683  | 0.177  | 0.758  |
> | BEVFusion | UAV3D  | Res-101  |  800*450  |  0.536 (+0.025)   |  0.582(+0.090)   |  0.521(+0.162)   |  0.154  (+0.023)   |   0.343 (+0.415) |
> | DETR3D|  UAV3D-200 | Res-101 |  800*450  |  0.461 |  0.525 | 0.750 | 0.179 | 0.172  |
> | DETR3D|   UAV3D |  Res-101 |  800*450   |  0.618 (+0.157)  |  0.671 (+0.146)  | 0.494 (+0.256)  | 0.158(+0.021)  |0.070(+0.102)   |

---

> > ### Comment · Reviewer_xNiq · 2024-08-26
> >
> > Thanks for the response and additional results. Most of my concerns have been addressed, although I think you should expand more on the sim2real part and similar comparisons between UAV3D-200 and UAV3D should be added to all experiments. I'll raise my score for your efforts on developing large-scale and open-source datasets.

---

### Official Review · Reviewer_LYae · 2024-07-24

**Rating:** 6
**Confidence:** 4
**Correctness:** The dataset is constructed soundly.
**Clarity:** The paper is overall well-written. So…

**Review:**

This paper introduces the UAV3D dataset, addressing the benchmark for cooperative perception with connected drones. Using the CARLA simulator, the dataset generates camera data for five drones and benchmarks various tasks. The paper is well-structured, clearly describes the data collection process, and provides comprehensive benchmarking of different methods across multiple tasks. However, I have several concerns:

1. Previous work, specifically CoPerception-UAV, has already generated a cooperative perception dataset for drones using CARLA. This limits the novelty of the current work. I'm not saying that a second dataset is without merit. As the authors point out, UAV3D is significantly larger than CoPerception-UAV, so its potential contribution to the community should not be overlooked. Additionally, synthetic datasets inherently have limitations for real-world applications

2. The paper mentions semantic segmentation labels. Semantic segmentation is highly valuable for research, and generating these labels in simulation is particularly useful since they are usually very costly to obtain. However, the paper does not benchmark this task, leaving the dataset's utility for semantic segmentation unassessed.

3. The paper mentions providing 27 different vehicle categories. Why vehicles specifically? It would be helpful for the authors to list these categories and explain how they differ from those in CoPerception-UAV.

4. In line 134, it's unclear why a 3D voxel grid is considered. If I understand correctly, the authors use BEV (Bird's Eye View) to represent 3D detection, meaning the information in the z-direction is compressed, so a 3D voxel grid might not be necessary. This needs clarification to avoid confusion.

5. The upper-bound used in the benchmarking seems to rely on raw data fusion, but the specific method is not explained. Did the authors use a specific early fusion approach, or was object detection performed directly using ground truth data?

6. Figure 4 does not effectively showcase the differences in performance of various algorithms on this dataset. Consider enlarging or highlighting the distinctions to make them clearer for reading.

**Strengths:**

1. As far as I know, the dataset provided in this article is the largest UAV dataset available, supporting both multi-agent and multi-view tasks. Although it is not the first of its kind, it should have significant potential value for the community.
2. This article not only benchmarks the 3D object detection task but also includes tracking tasks, making the benchmark results more comprehensive.
3. The article considers various methods and provides detailed descriptions of the models used.
4. The benchmarking uses multiple evaluation metrics, making the evaluation criteria more multidimensional.
5. Both the dataset and the code are accessible.

**Additional Feedback:**

I noticed that the baseline code appears to clone the original repository, including README-file. Does this mean no additional adaptation is required to use it with the authors' dataset? I haven't personally verified this, so I hope the authors can offer a simple explanation.

**Documentation:**

The authors have provided a project page that includes the dataset and the code used for benchmarking.

**Ethics:**

There are no immediate ethical concerns with the dataset.

**Limitations:**

The paper does not include a necessary discussion of its limitations. One major limitation is that the dataset is synthetic, which significantly restricts the evaluation of different methods in real-world applications. Although the authors mention future work involving data under various weather and lighting conditions, I believe collecting and creating real-world data may be more important.

**Opportunities For Improvement:**

1. This paper could benefit from a more detailed discussion comparing UAV3D with existing datasets, particularly CoPerception-UAV. For instance, comparisons could be made regarding sensor configurations, the number and flight patterns of drones, and the types of labels provided.
2. It would be useful to include some basic benchmarks for semantic segmentation tasks.
3. Some details require more thorough explanations, such as the vehicle categories, the use of the 3D voxel grid, and the specifics of the upper-bound method.
4. The results presented in Figure 4 need to be improved to better highlight the benchmark outcomes for the readers.

**Relation To Prior Work:**

Although the paper discusses previous work, I believe the authors should further highlight the differences between this dataset and previous similar works. These differences could include sensor configurations, the number of drones and their flight patterns, and the types of labels provided.

In addition, I notice that the authors mention some very recent work on cooperative perception, such as RCooper, which is commendable. There are also several other new datasets and methods that, while not necessary for quantitative comparison or benchmarking due to their recency, could be included in the related work section to further strengthen it, such as TUMTraf-V2X [1], HoloVIC [2], CoHFF [3], V2X-Real [4].

[1] Zimmer, Walter, et al. "TUMTraf V2X Cooperative Perception Dataset." CVPR. 2024.
[2] Ma, Cong, et al. "HoloVIC: Large-scale Dataset and Benchmark for Multi-Sensor Holographic Intersection and Vehicle-Infrastructure Cooperative." CVPR. 2024.
[3] Song, Rui, et al. "Collaborative Semantic Occupancy Prediction with Hybrid Feature Fusion in Connected Automated Vehicles." CVPR. 2024.
[4] Xiang, Hao, et al. "V2X-Real: a Largs-Scale Dataset for Vehicle-to-Everything Cooperative Perception." arXiv preprint arXiv:2403.16034. 2024.

**Summary And Contributions:**

This paper primarily introduces UAV3D, a new cooperative perception dataset designed for UAM. The dataset leverages CARLA simulation by installing cameras in five directions on virtual drones to collect synthetic RGB data. Four tasks were selected for benchmarking the dataset, covering 3D detection and tracking by both single drones and multiple drones in cooperation. These tasks were evaluated using seven metrics, providing some fundamental experimental results. Because scenarios involving drones are naturally well-suited for multi-agent tasks, cooperative perception using multiple drones is particularly significant. Existing related datasets are only few, as the authors mention, CoPerception-UAV is a relatively well-known dataset. However, UAV3D surpasses CoPerception-UAV in terms of scale, which is a major contribution of this paper. Additionally, the dataset adopts a format similar to NuScenes, making it easier for users to adapt their methods for benchmarking within this dataset, thereby enhancing its compatibility.

---

> ### Author Rebuttal · Authors · 2024-08-16
>
> Our project website has been transferred to GitHub: https://huiyegit.github.io/UAV3D_Benchmark/.
>
> **Concerns in review:**
>
> 1. Previous work, specifically CoPerception-UAV...... (Please see it in the original review.)
>
> Thanks for raising these questions. Table 1(**attached pdf file "tables.pdf" in rebuttal to reviewer #2**) summarizes the differences between CoPercpetion-UAVs and UAV3D. CoPerception-UAVs is a pioneer dataset in cooperative perception from the UAV platform. However, there is currently no publicly available **source code** to utilize this dataset. 1) The official where2comm repo does not support this dataset. Please refer to the official repo: https://github.com/MediaBrain-SJTU/Where2comm. 2) The authors did not reply to the source code issue. Please refer to the issue 1 and 5 : https://github.com/MediaBrain-SJTU/Where2comm/issues. To the best of our knowledge, our work is the first that provides the source code and dataset for large-scale 3D perception tasks.
>
>
> As for the limitations of synthetic dataset, we agree that synthetic dataset is limited for real-world applications. There is a domain gap between the real and simulated datasets, where the performance would degrade when applying  models trained with simulated data to the real-world applications. The typical approach to mitigate the domain gap is to apply the Domain Adaptation, which trains the model on the simulated dataset and  adapt the model to real-world applications.
> However, collecting a real-world large-scale 3D perception dataset for UAVs is extremely challenging due to security and privacy concerns, as well as the high costs of data labeling. Therefore, it is essential to have a synthetic dataset to advance research in 3D perception for UAVs until a real dataset becomes available.
>
> We will include these discussions and limitations in the paper.
>
>
> 2. The paper mentions semantic segmentation labels.......(Please see it in the original review.)
>
> Thanks for the suggestion. Besides object detection and tracking, semantic segmentation is another important perception task. The three baseline models:BEVFusion, DETR3D, and PETR, are specifically proposed for 3D object detection. Although our UAV3D includes semantic segmentation labels for RGB images, we did not evaluate this task in our current study. We will explore this in future work.
>
> 3. The paper mentions providing 27 different vehicle categories....... (Please see it in the original review.)
>
> Thanks for pointing out this issue. When collecting the UAV3D dataset, we spawned 200 vehicles of 27 different categories into the CARLA simulation environment. However, after reviewing the actual dataset, we found that only 17 vehicle categories are present in UAV3D. As shown in the figure(**attached pdf file "category.pdf" in rebuttal to all reviewers**), the categories are almost evenly distributed, as they were generated randomly by the simulation environment. In our benchmark experiments, we treated all vehicles as a single ``car" class for the perception task. However, the 17 categories are available for more fine-grained perception tasks.
>
> We reviewed the "category.json" file in CoPerception-UAVs and found that there are 21 vehicle classes. The comparison of vehicles in file “category.json" is listed in Table 2(**attached pdf file "tables.pdf" in rebuttal to reviewer #2**).
> We will clarify this vehicle class issue in the paper.
>
>
> 4. In line 134, it's unclear why a 3D voxel grid is considered......... (Please see it in the original review.)
>
> Thanks for pointing out this issue.
> In baseline BEVFusion, it lifts each feature pixel into D discrete points along the camera ray to create a camera feature point cloud of size N × H × W × D,  where N is the number of cameras and (H, W ) is the camera feature map size.The BEV pooling operation is performed to aggregate all features within each 128 × 128 BEV grid and flatten the features along the z-axis.
> We will clarify this in the paper.
>
> 5. The upper-bound used in the benchmarking seems to rely on raw data fusion.......(Please see it in the original review.)
>
> Thanks for pointing out this issue. The upper-bound baseline employs an early fusion strategy that aggregates raw images from all agents. While this approach fully leverages the available perception data, it requires a large amount of transmission bandwidth. As for the implementation of the upper-bound baseline, the 25 images from 5 drones are directly utilized by the BEVFusion model.
> We will clarify the implementation of the upper-bound baseline in the paper.
>
> 6. Figure 4 does not effectively showcase the differences in performance of various algorithms......(Please see it in the original review.)
>
> Thank you for the suggestion. To present the detection results of the six collaborative approaches, we visualized 30 images in a single figure, which is difficult to distinguish the differences. We will include five enlarged/highlighted figures, each containing six images, in the appendix to better visualize the results.
>
> **Relation To Prior Work:**
>
>  TUMTraf-V2X, HoloVIC, CoHFF, V2X-Real...... (Please see it in the original review.)
>
> Thanks for pointing out these recent works. TUMTraf-V2X, HoloVIC, CoHFF, and V2X-Real are proposed for the cooperative perception in autonomous driving, which are related to our UAV3D. We will update the related work section and include a discussion of them.
>
> **Additional Feedback:**
>
> I noticed that the baseline code appears to clone the original repository, including README-file...... (Please see it in the original review.)
>
> Thanks for pointing out this issue. The original source codes for PETR, BEVFusion, and DETR3D cannot be directly applied to run UAV3D. We need to modify their source codes in many places such as data input, modules of model architecture, and results evaluation. We just updated our Git repository with detailed instructions. Please refer to  the website: https://github.com/huiyegit/UAV3D.

---

> > ### Comment · Reviewer_LYae · 2024-08-21
> > **Two further questions**
> >
> > Thank you for the response. I have two further questions based on the reply.
> >
> > 1. The author mentioned that CoPerception-UAVs did not provide the source code for the dataset, while UAV3D did. Could you please clarify what kind of source code is being referred to? Is it the source code for generating the data, or are there API functions that might be needed to use the data? Additionally, could you point out the specific location of this code in the repository?
> >
> > 2. Furthermore, regarding the statement: "The original source codes for PETR, BEVFusion, and DETR3D cannot be directly applied to run UAV3D." Could you briefly explain what modifications were made to the original source codes to adapt them for UAV3D?

---

> ### Author Rebuttal · Authors · 2024-08-23
>
> **1.** The author mentioned that CoPerception-UAVs did not provide the source code for the dataset, while UAV3D did. Could you please clarify what kind of source code is being referred to? Is it the source code for generating the data, or are there API functions that might be needed to use the data? Additionally, could you point out the specific location of this code in the repository?
>
> Thanks for the follow-up questions. The source code refers to following two groups of code.
> **1) Source code for raw data preprocessing:** The raw data needs to be converted into a standard format suitable for model input. However, the source code designed to process one type of raw data (e.g., NuScenes) cannot be directly applied to another (e.g., UAV3D) as the definitions of the raw data (such as the 3D coordinate systems and 3D box annotations) differ.
> The source code for UAV3D ("uav_converter.py") is available in our GitHub repository: https://github.com/huiyegit/UAV3D/tree/main/perception/bevfusion/tools/data_converter
> **2) Source code of baseline models to run on UAV3D:** We need to modify the source code of the existing algorithms in several places in order to apply them on UAV3D, such as data input, modules of model architecture, and results evaluation.
> For example, in baseline BEVFusion, the python file for loading data ("nuscenes_dataset.py"):  https://github.com/huiyegit/UAV3D/blob/main/perception/bevfusion/mmdet3d/datasets/nuscenes_dataset.py, the file for 2D-3D projection of image features ("base.py"): https://github.com/huiyegit/UAV3D/blob/main/perception/bevfusion/mmdet3d/models/vtransforms/base.py, the files for evaluation ("evaluate.py", "loaders.py", "mean_ap.py", "postprocess.py" and "splits.py"): https://github.com/huiyegit/UAV3D/tree/main/perception/bevfusion/projects.
>
> **2.** Furthermore, regarding the statement: "The original source codes for PETR, BEVFusion, and DETR3D cannot be directly applied to run UAV3D." Could you briefly explain what modifications were made to the original source codes to adapt them for UAV3D?
>
> Thanks for the follow-up questions. The NuScenes dataset uses the global coordinate system based on the World Geodetic System 1984 (WGS-1984), where x-axis is the East-West direction, y-axis is the North-South direction, and z-axis is Up-Down direction. CARLA, which is used for UAV3D data collection, uses the Unreal Engine (UE) coordinate system, with
> x-forward, y-right, and z-up, returning coordinates in local space, while AirSim, on the other hand, adopts the North-East-Down (NED) coordinate system, where north aligns with the Unreal Engine x-axis.
>
> To run the baseline models, PETR, BEVFusion, and DETR3D, on UAV3D, the coordinate transformations must be performed during data loading, 2D-3D projection of image features within the model architecture, and evaluation of the results. These modifications need to be applied separately to each baseline model.

---

### Official Review · Reviewer_7jif · 2024-07-29
**3D Perception Synthesized Dataset for Unmanned Aerial Vehicles**

**Rating:** 6
**Confidence:** 4
**Correctness:** Yes.
**Clarity:** Yes, the paper is easy to follow.

**Review:**

**Strength:**

1. The paper is well written and easy to follow.
2. The authors propose the largest UAV dataset available, supporting both multi-agent and multi-view tasks.
3. The experiments were sufficient, and the authors created benchmarks for both detection and tracking.

**Weakness:**

1. Previous work like CoPerception-UAV has created a cooperative perception dataset for drones using CARLA. The authors should clarify how their approach differs from other methods, beyond just data quantity.
2. The dataset and experiments require clearer descriptions, including a complete list of the 27 vehicle classes, bar graphs showing the counts for each category in training and test sets, and specific implementation details for the Upper-bound in Tables 8 and 9.

**Strengths:**

See `Review`.

**Additional Feedback:**

See detailed comments in `Opportunities For Improvement`.

**Documentation:**

The authors  provide a link to access their code and dataset.

**Ethics:**

No.

**Limitations:**

The authors have discussed the limitations and future work in the Conclusion section. Besides, the primary limitation of the article is that disparities between the synthesized dataset and the real dataset will restrict the dataset's practical application

**Opportunities For Improvement:**

See `Review Weakness`. Besides, there are few other suggestions:

1. Comparing simulated and real datasets will enhance the article's practical applications.
1. The benchmarks include older methods as baselines, with the option to add newer methods.

**Relation To Prior Work:**

The authors offer a comparison of other dataset. And the authors should explicitly highlight how this dataset differs from similar studies beyond its scale.

**Summary And Contributions:**

The authors introduce UAV3D, a dataset for single-agent and collaborative 3D perception in UAVs, generated using the CARLA environment alongside AirSim for UAV simulation. UAV3D comprises 1,000 scenes, each of which has 20 frames and fully annotated with 3D bounding boxes on vehicles. The authors  conduct benchmarks for four 3D perception tasks: single UAV 3D object detection, single UAV object tracking, collaborative UAVs 3D object detection, and collaborative UAVs object tracking.

---

> ### Author Rebuttal · Authors · 2024-08-16
>
> Our project website has been transferred to GitHub: https://huiyegit.github.io/UAV3D_Benchmark/.
>
>
> **Weaknesses:**
> 1. Previous work like CoPerception-UAV has created a cooperative perception dataset for drones using CARLA. The authors should clarify how their approach differs from other methods, beyond just data quantity.
>
> Thanks for raising this question. Table 1 below summarizes the differences between CoPercpetion-UAVs and UAV3D. CoPerception-UAVs is a pioneer dataset in cooperative perception from the UAV platform. However, there is currently no publicly available **source code** to utilize this dataset. 1) The official where2comm repo does not support this dataset. Please refer to the official repo: https://github.com/MediaBrain-SJTU/Where2comm. 2) The authors did not reply to the source code issue. Please refer to the issue 1 and 5 : https://github.com/MediaBrain-SJTU/Where2comm/issues. To the best of our knowledge, our work is the first that provides the source code and dataset for large-scale 3D perception tasks. Please refer to our official repo : https://github.com/huiyegit/UAV3D.
> We will include this discussion in the paper.
>
>  .......................Table 1: Comparison between CoPerception-UAVs and UAV3D. .............
> |       | ....CoPerception-UAVs.... |    ............................UAV3D.............................  |
> | :--: | :---------------: | :--------------: |
> | Simulation |  Carla & AirSim | Carla & AirSim |
> | Maps         | Town 3, 4 &6    | Town 3, 6, 7&10 |
> | Agents       | 5 drones           | 5 drones |
> | Sensors     | 25 RGB cameras | 25 RGB cameras |
> | Height        | 40,60,80m           | 60m |
> | Formation  | Static V shape     | Static cross-shape |
> | #Categories | 21                      | 17 |
> | Frequency   | 0.25Hz               | 0.5Hz |
> | Labels         | 3D boxes & Semantic labels         |  3D boxes & Semantic labels   & Box velocity |
> | #Samples    | 5,276                 | 20,000 |
> | #Images      | 131,900              | 500,000 |
> | source code | ×                        | ✓ |
>
>
> 2. The dataset and experiments require clearer descriptions, including a complete list of the 27 vehicle classes, bar graphs showing the counts for each category in training and test sets, and specific implementation details for the Upper-bound in Tables 8 and 9.
>
>
> Thanks for pointing out this issue. When collecting the UAV3D dataset, we spawned 200 vehicles of 27 different categories into the CARLA simulation environment. However, after reviewing the actual dataset, we found that only 17 vehicle categories are present in UAV3D. As shown in the figure( **attached pdf file "category.pdf" in rebuttal to all reviewers**), the categories are almost evenly distributed, as they were generated randomly by the simulation environment. In our benchmark experiments, we treated all vehicles as a single ``car" class for the perception task. The 17 categories are available for more fine-grained perception tasks. But they are not explored in our benchmark experiments.
>
> As for the upper-bound baseline, it employs an early fusion strategy that aggregates raw images from all agents. While this approach fully leverages the available perception data, it requires a large amount of transmission bandwidth. Specifically, for the implementation of the upper-bound baseline, the 25 images from 5 drones are directly utilized by the BEVFusion model.
>
> We will clarify the vehicle class issue and the implementation of the upper-bound baseline in the paper.
>
>
> **Opportunities For Improvement**
>
> 1. Comparing simulated and real datasets will enhance the article's practical applications.
>
> Thank you for the suggestion. Real-world datasets typically involve more complex conditions, including localization errors, sensor synchronization issues, and communication latency between agents. In contrast, simulated datasets benefit from the controlled, ideal settings of a simulation environment. For example, the object category in NuScenes follows a long-tail distribution, whereas in UAV3D, it follows an even distribution.
>
> Apparently, there is a domain gap between the real and simulated datasets, where the performance would degrade when applying  models trained on simulated data to the real-world applications. The typical approach to mitigate the domain gap is to apply the Domain Adaptation, which trains the model on the simulated dataset and adapts the model to real-world applications.
> We will add the lack of real dataset comparison as one of the limitations of our work.
>
> 2. The benchmarks include older methods as baselines, with the option to add newer methods.
>
> Thank you for this suggestion. To evaluate the perception tasks on UAV3D, we selected three well-known baseline models from the autonomous driving community: BEVFusion, DETR3D, and PETR.
> However, the original source codes of them cannot be directly applied to run UAV3D. We need to modify their source codes in many places such as data input, modules of model architecture, and results evaluation. We just updated our Git repository with detailed instructions, which can be used to add newer methods. Please refer to  the website: https://github.com/huiyegit/UAV3D.

---

> > ### Comment · Reviewer_7jif · 2024-08-19
> >
> > Thanks to the author's response, I have the following further questions:
> >
> > 1. I did not find a description of the 17 categories in the original paper (only 27), does this indicate some problem with the presentation in the paper? Additionally, the $\dagger$ symbols in Tab. 1 lack clear meaning.
> >
> > 2. Why does PETR perform poorly on Res-101 in Tables 2 and 3? This is confusing.
> >
> > 3. The original paper should include key experimental settings, such as the number of queries and the size of the bev feature. It should also report the training overhead (GPU hours and memory usage) and inference speed of these baselines on the dataset.
> >
> > 4. The simplest method DETR3D outperforms PETR and BEVFusion, which contrasts with findings on the autonomous driving dataset. A more detailed analysis explaining why DETR3D excels is necessary.
> >
> > 5. UAV3D achieves significantly higher performance than NuScenes on NDS. Does this suggest that UAV3D is easier? The authors should discuss the challenges of the UAV3D dataset extensively and illustrate flaws in existing methods to justify further research.

---

> > > ### Author Rebuttal · Authors · 2024-08-21
> > >
> > > **1. I did not find a description of the 17 categories in the original paper (only 27), does this indicate some problem with the presentation in the paper? Additionally, the symbols in Tab. 1
> > > lack clear meaning.**
> > >
> > > Thanks for raising these questions. Yes. The value "27" represents the configuration setting in the CARLA simulation environment, while "17" corresponds to the actual vehicle classes in UAV3D. We will update the paper to fix this issue.
> > >
> > > The symbol &dagger;  denotes that the 17 categories can be used for more fine-grained perception tasks, while in our experiments, we treated all vehicles as a single ``car" class for the perception task. We will clarify this in the paper.
> > >
> > > **2.  Why does PETR perform poorly on Res-101 in Tables 2 and 3? This is confusing.**
> > >
> > > Thanks for raising this question. The original PETR GitHub repository provides a ResNet-50 configuration file with numerous hyperparameters for NuScenes. We adapted this configuration and achieved a good performance with the PETR baseline on UAV3D. Subsequently, we modified the backbone from ResNet-50 to ResNet-101 and adjusted some hyperparameters  to train on UAV3D. But the performance was worse than with ResNet-50. We did not include this unexpected result of PETR ResNet-101 in the paper as it was supposed to perform better than ResNet-50. To achieve better performance with ResNet-101, it would require significant time and effort to adjust several hyperparameters, including those for the model backbone, model neck, optimizer, and learning strategy. We will clarify the performance issue of PETR with ResNet-101 in the paper.
> > >
> > > **3. The original paper should include key experimental settings, such as the number of queries and the size of the bev feature. It should also report the training overhead (GPU hours and memory usage) and inference speed of these baselines on the dataset.**
> > >
> > > Thank you for the suggestions. We will provide the experimental settings, training overhead, and inference speed in the revised paper.
> > > Regarding the experimental settings, we aimed to maintain the configurations used in the original baseline models, PETR, BEVFusion, and DETR3D, while making the necessary adjustments to adapt them to the UAV3D dataset. Furthermore, to ensure a fair comparison among the three baseline models, we unified certain aspects of their architectures. For instance, while PETR, BEVFusion, and DETR3D originally use different backbones (ResNet-50, Swin-Transformer, and ResNet-101, respectively), we standardized the use of ResNet-50 and ResNet-101 as the model backbones across all three baselines.
> > >
> > > Regarding the experimental details, we have uploaded the training logs to the Git repository(https://github.com/huiyegit/UAV3D), which include information about the training process, such as loss values, learning rates and training time. We plan to create a table to report the GPU hours, memory usage, and inference speed, and we will add this table in the appendix.
> > >
> > > **4. The simplest method DETR3D outperforms PETR and BEVFusion, which contrasts with findings on the autonomous driving dataset. A more detailed analysis explaining why DETR3D excels is necessary.**
> > >
> > > Thank you for the suggestion. The three baseline models, PETR, BEVFusion, and DETR3D, were specifically designed for 3D object detection on the NuScenes dataset. The performances of these models on NuScenes are close, with a performance gap of about 2\% in the NDS metric. However, in our benchmark experiments on UAV3D, the performance gap is about 9\% in the NDS metric.
> > >
> > > All three models have a similar detection architecture, including a backbone, neck, and head. They all utilize the ResNet-50 backbone and employ similar neck architectures with minor variations: CPFPN for PETR, SecondFPN for BEVFusion, and FPN for DETR3D. The primary difference lies in their detection heads.
> > > PETR generates 3D position-aware features by encoding 3D positional embeddings into 2D image features. BEVFusion lifts each feature pixel along the camera ray to create a 3D feature point cloud. DETR3D transforms 2D features to refine object queries by projecting the 3D reference points into the image space. The 2D-to-3D feature transformation in DETR3D benefits significantly from the precise coordinates of vehicles and cameras from the simulation environment.
> > > We will include an analysis of these experimental results in the paper.
> > >
> > > **5.  UAV3D achieves significantly higher performance than NuScenes on NDS. Does this suggest that UAV3D is easier? The authors should discuss the challenges of the UAV3D dataset extensively and illustrate flaws in existing methods to justify further research.**
> > >
> > > Thanks for the suggestion. Datasets for autonomous driving (i.e.NuScenes) capture scenes from a ground-level perspective, similar to what a human driver would see, where objects would appear more occluded due to the limited perspective. On the other hand, UAV3D has a large view and perspective (bird's-eye), capturing images from a higher altitude. Therefore, the view and perspective is more global. We believe this global view of UAV3D is one reason why the detection performance is higher on UAV3D than on NuScenes. Furthermore,  the annotations are almost perfectly accurate on UAV3D, which can lead to high performance on test sets from the same simulation environment.
> > >
> > > Regarding the challenges in UAV3D, targets/objects in UAV3D often appear smaller and with fewer details. Although the benchmark baselines, PETR, BEVFusion and DETR3D, have demonstrated strong performance on UAV3D, they were originally designed for 3D object detection in autonomous driving and do not specifically address the challenges of UAV3D. A potential future research direction is to deal with detecting small objects from high altitudes and addressing occlusions caused by buildings or other structures. We believe there is still a large room to improve perception performance on UAV3D. We will include these discussions in the paper.

---

> > > > ### Comment · Reviewer_7jif · 2024-08-21
> > > >
> > > > Thanks to the authors for their responses. Does ResNet101 utilize DCN, and what pre-training weights are used for Res50 and Res101? Regarding the detector mentioned in the article, is the BEV feature's coordinate system based on the UAV's ego coordinate system or the ground? How is the 3D coordinate of PETR defined?

---

> > > > > ### Author Rebuttal · Authors · 2024-08-23
> > > > >
> > > > > Thanks for the follow-up questions. Yes, in the config file of the backbone ResNet-101, the DCN version 2 is used in the last two (out of four) training stages, allowing the model to adapt its receptive fields more flexibly in these deeper layers. In PETR, the pre-trained ResNet-50 and ResNet-101 are the "caffe" version. The setting "frozen\_stages=-1" means that no stages are frozen. In other words, all stages of the ResNet backbone are trainable, and their weights will be updated during the training process.
> > > > >
> > > > > There are three types of coordinate system: sensor (camera) coordinate system, ego (UAVs) coordinate system and global coordinate system. The BEV's image features are projected to ego coordinate system with the origin of z-axis on the ground.
> > > > >
> > > > > PETR has designed a 3D coordinates generator, where the camera frustum space (shared by all camera views) is discretized into a 3D meshgrid. The meshgrid coordinates are transformed by different camera parameters, resulting in the coordinates in 3D world space.
> > > > >
> > > > > We will clarify these details in the appendix.

---

> > > > > > ### Comment · Reviewer_7jif · 2024-08-25
> > > > > >
> > > > > > Most of my concerns have been addressed. The author's experimental settings primarily reference the nuScense dataset, but more recent baseline validations are lacking. Given the scale and public availability of the dataset discussed in the article, I will keep the positive rating.

---

### Author Rebuttal · Authors · 2024-08-16

Our project website has been transferred to GitHub:https://huiyegit.github.io/UAV3D_Benchmark/.

Please find the attached file "category.pdf"which includes the bar graph of vehicle categories.

---

### Decision · Program_Chairs · 2024-09-26

**Decision:**

Accept (Poster)

**Comment:**

This paper introduces a synthesized dataset for single-agent and collaborative 3D perception in UAVs, primarily referencing the nuScenes dataset for its experimental settings. All reviewers rated it positively, though there are issues to address, such as expanding on sim2real aspects, comparing UAV3D-200 with UAV3D, validating against advanced baselines, and conducting further dataset analysis. Given its scale and availability, the dataset has potential impact. The AC recommends accepting this paper, hoping the authors will refine and revise it based on the discussions during the rebuttal.